# Less-Forgetting Multi-Lingual Fine-Tuning

**Yuren Mao**[1]  **Yaobo Liang**[2*]  **Nan Duan**[2]  **Haobo Wang**[1]
**Kai Wang**[3]  **Lu Chen**[1]  **Yunjun Gao**[1*]

[1]Zhejiang University
[2]Microsoft Research Asia
[3]Shanghai Jiao Tong University.
{yuren.mao,wanghaobo,luchen,gaoyj}@zju.edu.cn
{yalia,nanduan}@microsoft.com, w.kai@sjtu.edu.cn

## Abstract

Multi-lingual fine-tuning (MLF), which fine-tunes a multi-lingual language model (MLLM) with multiple source languages, aims to gain good zero-shot performance on target languages. In MLF, the fine-tuned model tends to fit the source languages while forgetting its cross-lingual knowledge obtained from the pre-training stage. This forgetting phenomenon degenerates the zero-shot performance of MLF, which remains under-explored. To fill this gap, this paper proposes a multi-lingual fine-tuning method, dubbed Less-forgetting Multi-lingual Fine-tuning (LF-MLF). In LF-MLF, we cast multi-lingual fine-tuning as a constrained optimization problem, where the optimization objective is to minimize forgetting, and constraints are reducing the fine-tuning loss. The proposed method has superior zero-shot performance; furthermore, it can achieve the Pareto stationarity. Extensive experiments on Named Entity Recognition, Question Answering and Natural Language Inference back up our theoretical analysis and validate the superiority of our proposals.

## 1 Introduction

Multi-lingual Language Models (MLLMs), which are pre-trained on multiple languages, have the cross-lingual generalization ability. By fine-tuning MLLMs on source languages, we can expect promising zero-shot performance on target languages [1, 2]. In literature, most of the works focus on mono-lingual fine-tuning, which fine-tunes a multilingual model on only one source language (typically English). However, in practice, it is common that there are multiple source languages available, e.g., in name entity recognition, it is easy to obtain labeled data for English, German, French and so on. In this background, multi-lingual fine-tuning (MLF), which fine-tunes an MLLM on multiple source languages, becomes an emerging research topic.

In the fine-tuning procedure of MLF, each source language has a language-specific gradient descent direction. Combining these gradient descent directions together, we can find a set of gradient descent directions common for all the source languages. However, in this set, some directions lead a MLLM to fit the source languages too much and forget its cross-lingual knowledge obtained from the pre-training stage. This forgetting phenomenon can degenerate the zero-shot performance of MLF. Thus, *how to find a common gradient descent direction that benefits both the fine-tuning performance and zero-shot performance* is a important issue in MLF, which remains under-explored.

To address this issue, this paper proposes to find a *less-forgetting descent direction*, which prevents a MLLM from forgetting the cross-lingual generalization ability and is a common descent direction for the source languages. To find this less-forgetting descent direction, we cast MLF as a constrained

---

*Corresponding author.

36th Conference on Neural Information Processing Systems (NeurIPS 2022).

optimization problem. The optimization objective is to minimize the forgetting of a MLLM's cross-lingual generalization ability, and the constraint is that the direction should be a descent direction common to all the source languages. Correspondingly, we propose a novel algorithm, namely Less-forgetting Multi-lingual Fine-tuning (LF-MLF), to solve this constrained optimization problem and obtain the less-forgetting descent direction.

For LF-MLF, we have conducted both theoretical and experimental analysis. Theoretical results demonstrate that LF-MLF can effectively reduce the forgetting of multi-lingual models in fine-tuning; besides, LF-MLF can achieve the Pareto stationarity. The superiority of LF-MLF has been experimentally verified on three kind of tasks (i.e., Named Entity Recognition, Question Answering and Natural Language Inference). Furthermore, we also conduct experimental analysis on several factors that impact the performance of multi-lingual fine-tuning (e.g., the impact of the number of the fine-tuning languages), which have the potential to forge new trends in multi-lingual fine-tuning.

## 2 Related Works

**Multi-lingual Language Models (MLLMs)** are pre-trained using tons of unlabeled data from multiple languages and expected to own the cross-lingual zero-shot transfer ability. This ability can facilitate the low resource languages processing with knowledge transferred from the high resource languages [2]. Several MLLMs have been proposed, such as mBERT [3] and XLM-R [4]. Given a MLLM, most of the existing works analyze its cross-lingual zero-shot transfer ability by observing the model's zero-shot performance on the target languages after mono-lingual fine-tuning [5, 6, 7]. Few works [8, 9] have investigated the multi-lingual fine-tuning setting, but they do not consider the forgetting of MLLMs' cross-lingual generalization ability. To fill this gap, this paper focuses on the forgetting problem in multi-lingual fine-tuning.

**Continual Learning (CL)** studies the problem of learning a model on sequential tasks without forgetting knowledge obtained from the preceding tasks [10]. In CL, reducing forgetting is the major challenge. Lots of methods have been proposed to reduce forgetting, such as GEM [11], EWC [12] and IMM [13]; however, these methods typically focus on the single-objective optimization setting, where there is only one learning objective. None of them considers the multi-objective optimization setting, where multiple learning objectives involve. Nevertheless, multi-lingual fine-tuning involves a multi-objective optimization problem, which aims to optimize the fine-tuning performance on all the including languages. Thus, existing CL methods are not suitable for multi-lingual fine-tuning.

**Multi-task Learning (MTL)**, which simultaneously learns multiple tasks, aims to achieve proper performance on all included tasks. Recently, various MTL methods [14, 15, 16, 17, 18, 19, 20] have been proposed. Among them, PCGrad [19] and Gradient Vaccine [20] have achieved the state-of-the-art performance. Regarding the source languages of MLF as MTL tasks, MTL methods can be used in MLF. However, MTL methods just focus on the performance of including source languages (tasks), while they do not consider the performance of target languages. It will bring inferior zero-shot performance on the target languages. For example, PCGrad and Gradient Vaccine just focus on finding a proper common descent direction for the source languages, but the target languages are not considered in these methods. Thus, they cannot expect to achieve promising zero-shot performance. By contrast, the proposed LF-MLF finds the common descent direction that benefit the zero-shot performance. Overall, LF-MLF have more advantages over the MTL methods in MLF.

## 3 Less-forgetting Multi-lingual Fine-tuning

The goal of multi-lingual fine-tuning is twofold: firstly, achieving proper fine-tuning performance on the source languages, and secondly, achieving proper zero-shot performance on the target languages. For the first goal, it is necessary to avoid the model from forgetting the cross-lingual knowledge it obtained in the pre-training phase. For the second goal, we face a multi-objective optimization problem, where the losses for the source languages should be jointly optimized. In this section, we firstly give a formal definition of the multi-lingual fine-tuning and then propose a method (dubbed LF-MLF) that can simultaneously achieve the above two goals.

## 3.1 Problem Definition of Multi-lingual Fine-tuning

Given a multi-lingual language model $\theta_p$, which is pre-trained using large amounts of unlabeled data from language set $S_p$, we aim to fine-tune this model on a downstream task and achieve proper performance. For the downstream task, we have task-specific labeled training data from a set of source languages $S_f$, where $S_f \subseteq S_p$. After fine-tuning, $\theta_p$ has been updated as $\theta_f$. Assume there are $T$ languages in source languages set $S_f$. For each language, the task-specific loss is denoted as $L_t(\theta_f)$. Under the gradient descent paradigm of deep learning, in the $k^{th}$ iteration of multi-lingual fine-tuning, the model is updated by

$$\theta_k = \theta_{k-1} - \eta \sum_{t=1}^{T} w_t^k \nabla L_t(\theta_{k-1}). \tag{1}$$

where $\eta$ is the learning rate, $\sum_{t=1}^{T} w_t^k = 1$ and $w_t^k \geq 0, t \in \{1,...,T\}$. In Eq. (1), $-\sum_{t=1}^{T} w_t^k \nabla L_t(\theta_{k-1})$ is the convex hull of the gradient descent directions of the included languages, which contains feasible gradient descent directions for the multi-lingual fine-tuning.

Furthermore, denote the pre-training loss of $\theta_p$ and $\theta_f$ as $L_p(\theta_p)$ and $L_p(\theta_f)$ respectively. According to the classic definition of forgetting [21], we define the forgetting between $\theta_p$ and $\theta_f$ as $F_p = L_p(\theta_f) - L_p(\theta_p)$, which quantifies the performance degeneration of $\theta_f$ on the multi-lingual pre-training task (e.g., masking tokens). With the above notation, the two goals for multi-lingual fine-tuning can be formally defined as: (1) minimizing $F_p$, and (2) minimizing $\{L_1(\theta_f), ..., L_T(\theta_f)\}$.

## 3.2 Upper Bound the Forgetting in Multi-lingual Fine-tuning

In fine-tuning, we cannot directly compute the forgetting $F_p$. Alternatively, we propose to minimize the forgetting by minimizing its upper bound. In Theorem 1, we give a upper bound for the forgetting of multi-lingual fine-tuning. The assumptions in Theorem 1 are widely accepted [21, 22, 23] and can ensure proper performance.

**Theorem 1.** *Assume $L_p(\theta_f)$ can be approximated by its second order Taylor expansion and $\theta_p$ is a minima w.r.t the pre-training loss. Then, we have*

$$F_p \leq \frac{\lambda_p \eta}{2} \| \sum_{k=1}^{K} \sum_{t=1}^{T} w_t^k \nabla L_t(\theta_{k-1}) \|^2. \tag{2}$$

*where $\lambda_p$ is the maximum eigenvalue of $\nabla^2 L_p(\theta_p)$, and $K$ is the number of iterations of fine-tuning.*

## 3.3 Less-forgetting Multi-lingual Fine-tuning

In multi-lingual fine-tuning, we not only need to keep the multi-lingual memory of the pre-trained model but also need to achieve proper fine-tuning performance, i.e., minimizing $F_p$ and $\{L_1(\theta_f), ..., L_T(\theta_f)\}$ together. Achieving these two goals at same time is challenging and remains unexplored. In this section, we propose a novel method, dubbed Less-forgetting Multi-lingual Fine-tuning (LF-MLF), to jointly achieve these two goals.

Specifically, we minimizing $F_p$ by tightening its upper bound introduced in Theorem 1. Besides, minimizing $\{L_1(\theta_f), ..., L_T(\theta_f)\}$ is a multi-objective optimization (MOO) problem. We solve this MOO problem by the means of multi-objective gradient descent.

According to Theorem 1, in the $k^{th}$ iteration,

$$F_p^k \leq \frac{\lambda_p \eta}{2} \| \sum_{i=1}^{k} \sum_{t=1}^{T} w_t^i \nabla L_t(\theta_{i-1}) \|^2 = \frac{\lambda_p \eta}{2} \| \overline{\nabla} L + \sum_{t=1}^{T} w_t^k \nabla L_t(\theta_{k-1}) \|^2 \tag{3}$$

where $\overline{\nabla} L = \sum_{i=1}^{k-1} \sum_{t=1}^{T} w_t^i \nabla L_t(\theta_{i-1})$ and $\theta_0 = \theta_p$.

To reduce fogetting, in the $k^{th}$ iteration, we propose to minimize $F_p^k$'s upper bound as in Eq. (4).

$$\min_{w^k} \| \overline{\nabla} L + \sum_{t=1}^{T} w_t^k \nabla L_t(\theta_{k-1}) \|^2 \tag{4}$$

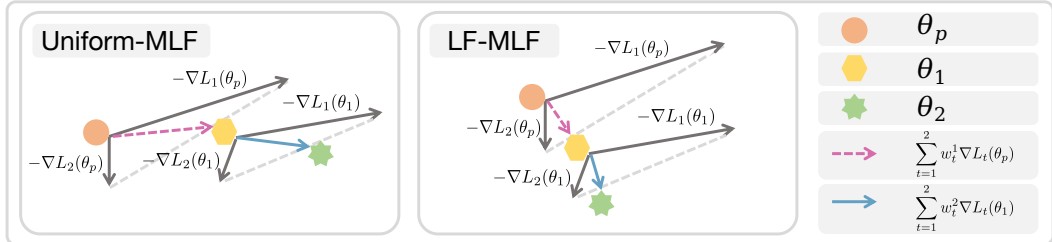

Figure 1: Toy example: comparison the model update procedures of LF-MLF and Uniform-MLF.

Except for reducing forgetting, we also need to decrease the task-specific losses of the fine-tuned languages, i.e, move toward the objective of minimizing $\{L_1(\theta_f), ..., L_T(\theta_f)\}$. Thus, in the $k^{th}$ iteration, we need to update the model in a descent direction common to all the source languages, which is the principle of multi-objective gradient descent. Under this principle, the update direction should not have a obtuse angle with the gradient direction on each language as in Eq. (5).

$$(\sum_{t=1}^{T} w_t^k \nabla L_t(\theta_{k-1}))^\top \nabla L_t(\theta_{k-1}) \geq 0, \ for \ t \in \{1, ..., T\}. \tag{5}$$

To reduce fogetting and minimizing the downstream task losses, we combine Eq. (4) with Eq. (5) and propose to formulate the less-forgetting multi-lingual fine-tuning as the following constrained optimization problem.

**Problem 1.**

$$\begin{aligned}
\min_{w^k} \quad & \|\bar{\nabla}L + \sum_{t=1}^{T} w_t^k \nabla L_t(\theta_{k-1})\|^2 \\
s.t. \quad & (\sum_{t=1}^{T} w_t^k \nabla L_t(\theta_{k-1}))^\top L_t(\theta_{k-1}) \geq 0, t \in \{1, ..., T\} \\
& \sum_{t=1}^{T} w_t^k = 1 \\
& w_t^k \geq 0, t \in \{1, ..., T\}.
\end{aligned} \tag{6}$$

To illustrate the rationale of this constrained optimization problem, we give a toy example in Figure 1. In this example, we compare LF-MLF with the Uniform-MLF method which adopts the uniform weights across the training procedure (i.e., $w_t^k = 1 for \ t \in \{1, ..., T\} \ and \ k \in \{1, ..., K\}$). LF-MLF chooses the common descent direction that minimizes the distance between the original MLLM $\theta_p$ and the updated MLLM $\theta_k$ at each update step. Obviously, LF-MLF has shorter distance than the Uniform-MLF method, which illustrates that LF-MLF can slow the forgetting of original MLLM down. Furthermore, to solve Problem 1, we formulate it as the following quadratic programming problem with $T$ variables and $2T + 1$ constraints.

**Problem 2.**

$$\begin{aligned}
\min_{w^k} \quad & \frac{1}{2}(w^k)^\top G w^k + (w^k)^\top c \\
s.t. \quad & b_t^\top w^k \geq 0, t \in \{1, ..., T\} \\
& \sum_{t=1}^{T} w_t^k = 1 \\
& w_t^k \geq 0, t \in \{1, ..., T\}.
\end{aligned} \tag{7}$$

where $G = \nabla L(\theta_{k-1})^T \nabla L(\theta_{k-1})$, $\nabla L(\theta_{k-1}) = [\nabla L_1(\theta_{k-1}), L_2(\theta_{k-1}), ..., L_T(\theta_{k-1})]^\top$. $c = \nabla L(\theta_{k-1})^T \bar{\nabla}L$ and $b_t = \nabla L(\theta_{k-1})^\top \nabla L_t(\theta_{k-1})$.

Problem 2 can be effectively solved. This paper adopts a interior-point solver [24]. Let $w_k^*$ be the solution of the above quadratic programming problem. In our proposed LF-MLF, the update rule is

$$\theta_k = \theta_{k-1} - \eta \sum_{t=1}^{T} (w_k^*)^t \nabla L_t(\theta_{k-1}). \tag{8}$$

Overall, the challenging multi-lingual fine-tuning problem is transferred to a simple quadratic programming problem. The detailed steps of LF-MLF can be referred to Algorithm 1.

---

**Algorithm 1:** Less-forgetting Multi-lingual Fine-tuning (LF-MLF)

---
**Input:** Pre-trained model $\theta_p$, Number of iterations $K$.
**for** $k \leftarrow 1$ **to** $K$ **do**
  **if** k == 1 **then**
    Solve Problem. (2) with $c = [0, 0, ..., 0]^\top$ and obtain solution $w_1^*$.
    $\theta_1 = \theta_p - \eta \sum_{t=1}^{T} (w_1^*)^t \nabla L_t(\theta_p)$.
  **else**
    Solve Problem. (2) and obtain solution $w_k^*$.
    $\theta_k = \theta_{k-1} - \eta \sum_{t=1}^{T} (w_k^*)^t \nabla L_t(\theta_{k-1})$
  **end if**
  **if** the feasible set of Problem. (2) is empty **then**
    break.
  **end if**
**end for**
**return** $\theta_K$.

---

## 4 Theoretical Analysis

In less-forgetting multi-lingual fine-tuning, we explicitly set the goal, minimizing the upper bound, as the optimization objective. Therefore, the less-forgetting property of the proposed LF-MLF method is straightforward. By contrast, the target of minimizing $\{L_1(\theta_f), ..., L_T(\theta_f)\}$ is achieved by the means of putting constraints on the update direction, which forces the update direction to be the common decreasing direction for all the fine-tuned languages. Its effectiveness seems not obvious. To make it clear, in this section, we conduct theoretical analysis on LF-MLF's fine-tuning performance.

In multi-lingual fine-tuning, we aim to jointly fine-tune several languages, namely minimizing the loss vector $\{L_1(\theta_f), ..., L_T(\theta_f)\}$. It is a multi-objective optimization problem, where the optimal solutions are Pareto optimal as in Definition 1.

**Definition 1** (Pareto Optimality). *For $\boldsymbol{L}(\theta) = [\{L_1(\theta), ..., L_T(\theta)\}]^\top$, the Pareto optimality w.r.t the optimization objective $\min_\theta \boldsymbol{L}(\theta)$ is defined as follows: (1) A solution $\theta$ dominates a solution $\overline{\theta}$ if $L_t(\theta) \leq L_t(\overline{\theta})$ for all $t \in T$ and $\boldsymbol{L}(\theta) \neq \boldsymbol{L}(\overline{\theta})$ and (2) A solution $\theta^*$ is deemed Pareto optimal if there exists no solution $\theta$ that dominates $\theta^*$.*

However, in deep learning, the parameter spaces are highly non-convex. To the best of our knowledge, there is no method that can learn a Pareto optimal deep neural network model. Settling for less, the best we can do is to achieve the Pareto stationarity as in Definition 2. Pareto stationarity is a necessary condition of achieving Pareto optimality.

**Definition 2** (Pareto Stationarity). *For $\boldsymbol{L}(\theta^0) = [\{L_1(\theta^0), ..., L_T(\theta^0)\}]^\top$ is said to be Pareto-stationary at the design-point $\theta^0$ iff there exists a convex combination of the gradient-vectors, $\sum_{t=1}^{T} w_t \nabla L_t(\theta^0)$, that is equal to zero:*

$$\sum_{t=1}^{T} w_t \nabla L_t(\theta^0) = 0; \quad \sum_{t=1}^{T} w_t^k = 1; \quad w_t^k \geq 0, \ t \in \{1, ..., T\}. \tag{9}$$

In Theorem 2 and 3, we propose that LF-MLF can achieve Pareto stationarity, which theoretically verifies that our proposed method can achieve proper performance on the fine-tuning languages.

**Theorem 2.** *Let $\mathcal{H}$ be a Hilbert space of finite dimension $N$. Let $L_t(\theta_k)$ $(1 \leq t \leq T \leq N)$ be $T$ smooth functions of the vector $\theta_k \in \mathcal{H}$, and $\theta_k^0$ a particular admissible design-point. Let $w_k^*$ be the solution of Problem 2 and descent direction $\nabla L = \sum_{t=1}^{T} (w_k^*)^t \nabla L_t(\theta_k)$. Then, either $\nabla L = \varnothing$, and $[L_1(\theta_k^0), ..., L_T(\theta_k^0)]^\top$ are pareto stationary at $\theta_k^0$ or $\nabla L \neq \varnothing$ and $-\nabla L$ is a descent direction common to all $\{L_t(\theta_k)\}_{t=1}^{T}$.*

**Theorem 3.** *LF-MLF can stop after a finite number of iterations if a Pareto stationary point is reached. Otherwise, If the sequence of iterates $\{\theta_k\}_{k=1}^{K}$ of the LF-MLF is infinite, it admits a weakly convergent subsequence.*

# 5  Experiments

In this section, we perform experimental studies on three downstream tasks– Named Entity Recognition (NER), Question Answering (QA) and Natural Language Inference (NLI) respectively to evaluate the performance of our proposed LF-MLF and verify our theoretical analysis.

## 5.1  Experimental Setup

**Datasets:** In our experiments, we adopt the NER [25], TyDiQA-GoldP [26] and XNLI [27] datasets for NER, QA and NLI respectively from the XTREME benchmark [8]. The details of these datasets are introuced in the supplementary material.

**Baselines:** We compare our proposed LF-MLF with the following baselines: (1) **Single-lingual Fine-tuning (SLF):** fine-tuning on each language independently; (2) **Uniform-MLF:** fine-tuning on the included languages simultaneously by means of a uniformly weighted sum of the language-specific gradient descent directions, i.e., $w_t^k = \frac{1}{T}$ constantly; (3) **PCGrad-MLF:** using the Project Conflicting Gradients (PCGrad) method proposed by [19] on multi-lingual fine-tuning; (4) **GradVac-MLF:** using the Gradient Vaccine (GradVac) method proposed by [20] on multi-lingual fine-tuning.

**Experimental Settings:** In our experiments, we generally fine-tune XLM-RoBERTa models (base-sized model) [28] with a training batch size of 32, and AdamW [29] is used as the optimizer. As to the numbers of fine-tuning epochs, we adopt the default setting of the XTREME benchmark, that are 10, 3, 5 for NER, TyDiQA and XNLI respectively. Besides, the learning rates are selected from $\{1e-5, 2e-5, 5e-5\}$, $\{1e-5, 2e-5, 5e-5\}$ and $\{5e-6, 1e-5, 2e-5\}$ with grid search for NER, TyDiQA and XNLI respectively. All the results are averaged over 3 runs.

## 5.2  Performance of Multi-lingual Fine-tuning

Conducting experiments on NER, TyDiQA and XNLI, we compare LF-MLF with the baselines on the zero-shot performance and fine-tuning performance respectively. For each dataset, we choose two or three source languages for fine-tuning according the following rules: (1) belongs to the Top 10 high-resource languages w.r.t Wikipedias; (2) the higher-resource the better. For NER and XNLI, *en, de, fr* are chosen, while *en, ru* are chosen for TyDiQA.

### 5.2.1  Zero-shot Performance

To evaluate the zero-shot performance of LF-MLF and the baselines, we record the F1 score performance (mean and standard deviation) over the target languages that have not been fine-tuned (i.e,. the other 45 languages except *en, de, fr* in NER, the other 7 languages except *en, ru* in TyDiQA and the other 10 languages except *en, de, fr* in XNLI). Furthermore, pairwise t-tests at 0.05 significance level are conducted based on the three runs. The results are reported in Table 1, 2 and 3 for NER, TyDiQA and XNLI respectively.

Besides, to comprehensively evaluate the superiority of LF-MLF, we further utilized Friedman test as the statistical test to analyze the relative performance among the compared methods across the three applications. At 0.05 significance level, the Friedman statistics is 5.5, and critical is 4.76. Thus, at 0.05 significance level,

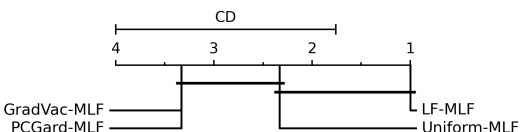

Figure 2: CD diagram of Bonferroni-Dunn test.

the null hypothesis of indistinguishable performance of LF-MTF among all compared methods is clearly rejected. Subsequently, we employ the Bonferroni-Dunn test as the post-hoc test by regarding LF-MLF as the control approach. Figure 2 reports the CD diagrams at 0.1 significance level, where the average ranks of the compared approaches is marked along the axis. From this figure, we can see that LF-MTF achieves highly superior results to other baseline methods.

### 5.2.2  Fine-tuning Performance

Except the zero-shot performance, we also care about the fine-tuning performance, namely the F1-score on the languages that have been fine-tuned. The results of fine-tuning performance are reported in Table 4, 5 and 6 for NER, TyDiQA and XNLI respectively. From these results, we can see that:

| Method | en-SLF | de-SLF | fr-SLF | Uniform-MLF | PCGrad-MLF | GradVac-MLF | LF-MLF |
|---|---|---|---|---|---|---|---|
| Zero-Shot | 60.05±0.18• | 58.53±0.19• | 63.97±0.58• | 67.45±0.35• | 67.40±0.32• | 67.57±0.25• | **68.50±0.21** |

Table 1: Zero-shot performance (F1 score (mean±std)) on NER. In addition, • indicates that LF-MLF is statistically superior to the comparing method (pairwise t-test at 0.05 significance level).

| Method | en-SLF | ru-SLF | Uniform-MLF | PCGrad-MLF | GradVac-MLF | LF-MLF |
|---|---|---|---|---|---|---|
| Zero-Shot | 55.05±0.67• | 60.66±0.34• | 62.51±0.43 | 61.74±0.51• | 61.36±0.36• | **63.03±0.31** |

Table 2: Zero-shot performance (F1 score (mean±std)) on TyDiQA. In addition, • indicates that LF-MLF is statistically superior to the comparing method (pairwise t-test at 0.05 significance level).

| Method | en-SLF | de-SLF | fr-SLF | Uniform-MLF | PCGrad-MLF | GradVac-MLF | LF-MLF |
|---|---|---|---|---|---|---|---|
| Zero-Shot | 72.46±0.39• | 74.26±0.38• | 74.01±0.14• | 75.17±0.12• | 74.87±0.21• | 74.67±0.14• | **75.56±0.15** |

Table 3: Zero-shot performance (F1 score (mean±std)) on XNLI. In addition, • indicates that LF-MLF is statistically superior to the comparing method (pairwise t-test at 0.05 significance level).

| Method | en-SLF | de-SLF | fr-SLF | Uniform-MLF | PCGrad-MLF | GradVac-MLF | LF-MLF |
|---|---|---|---|---|---|---|---|
| en | 82.59±0.14 | – | – | **83.30±0.15** | 81.76±0.08 | 82.07±0.18 | 82.75±0.10 |
| de | – | 88.78±0.09 | – | 87.90±0.03 | 87.13±0.05 | 87.30±0.13 | **88.92±0.09** |
| fr | – | – | **89.75±0.09** | 89.68±0.17 | 88.85±0.19 | 89.70±0.33 | 89.15±0.26 |

Table 4: Fine-tuning performance (F1 score (mean±std)) on NER.

| Method | en-SLF | ru-SLF | Uniform-MLF | PCGrad-MLF | GradVac-MLF | LF-MLF |
|---|---|---|---|---|---|---|
| en | 66.03±0.31 | – | 68.71±0.57 | 69.28±0.63 | 67.26±0.38 | **69.31±0.41** |
| ru | – | 70.89±0.48 | **72.43±0.08** | 71.76±0.19 | 71.87±0.15 | 71.66±0.21 |

Table 5: Fine-tuning performance (F1 score (mean±std)) on TyDiQA-GoldP.

| Method | en-SLF | de-SLF | fr-SLF | Uniform-MLF | PCGrad-MLF | GradVac-MLF | LF-MLF |
|---|---|---|---|---|---|---|---|
| en | 84.29±0.35 | – | – | **84.93±0.03** | 84.75±0.21 | 84.59±0.11 | 84.81±0.14 |
| de | – | 78.98±0.23 | – | 79.92±0.07 | 79.48±0.18 | 79.96±0.12 | **80.01±0.15** |
| fr | – | – | 79.80±0.36 | 80.63±0.11 | 79.98±0.12 | 80.43±0.06 | **80.83±0.13** |

Table 6: Fine-tuning performance (F1 score (mean±std)) on XNLI.

(1) the multi-lingual methods do not clearly outperform single-lingual methods on corresponding languages; (2) LF-MLF's performance is comparable to the baselines, and there is no baseline method can dominate LF-MLF. PCGrad-MLF and GradVac-MLF have inferior performance comparing with Uniform-MLF and LF-MLF. It because that PCGrad-MLF and GradVac-MLF cannot guarantee to find a Pareto stationary point, while Uniform-MLF and LF-MLF can find a Pareto stationary point. It verifies the theoretical analysis proposed in Section 4.

Overall, from section 5.2.1 and 5.2.2, we can see that LF-MLF has superior zero-shot performance and comparable fine-tuning performance. It verifies that LF-MLF has superior cross-lingual transfer ability. The superiority comes from LF-MLF's less-forgetting property, because such property enables LF-MLF to remember more about the target languages.

## 5.3  Impact of the Number of the Source Languages

In the last section, it is concluded that multi-lingual fine-tuning can significantly improve the zero-shot performance. Then, it is natural to ask *whether the zero-shot performance can be further improved when more languages have been fine-tuned*? In this section, we experimentally present that *the answer*

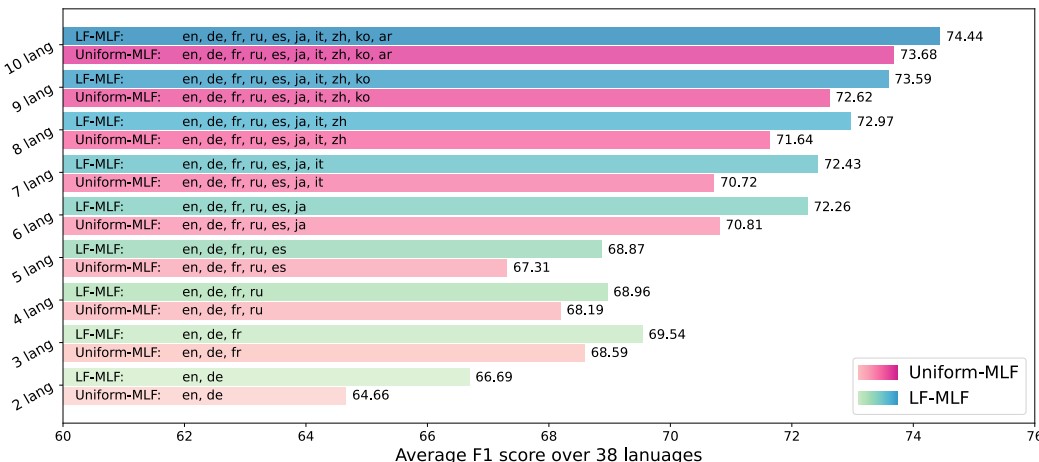

Figure 3: Zero-shot performance w.r.t different number of fine-tuning languages.

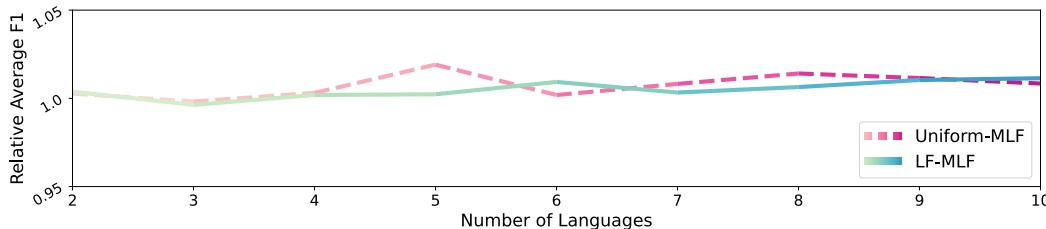

Figure 4: Relative fine-tuning performance w.r.t different number of fine-tuning languages.

*is yes*. Besides, we also investigate the impact of the number of source languages on the fine-tuning performance. Our experiments are conducted on the NER dataset, and the source languages are selected from the Top 10 high-resource languages w.r.t Wikipedias, i.e., *en, de, fr, ru, es, ja, it, zh, ko, ar*. By contrast, the other thirty eight languages are used as target languages. In the experiments, we increase the source languages from Top 2 to Top 10 and record the changing of zero-shot and fine-tuning performance.

### 5.3.1 Impact on the Zero-shot Performance

Figure 3 illustrates the change of MLF's zero-shot performance along with the increasing number of source languages. From this figure, we can conclude that (1) generally speaking, the zero-shot performance of a multi-lingual model continuously improves with the increasing of the number of source languages; (2) LF-MLF consistently outperforms Uniform-MLF when the number of source languages changes.

### 5.3.2 Impact on the Fine-tuning Performance

To evaluate the fine-tuning performance over different number of source languages, we present a metric, dubbed average relative F1 score, which measures the average relative improvement of multi-lingual methods's fine-tuning performance comparing with the single-lingual fine-tuning. Specifically, let $S_t^s$ be the F1 score of the single-lingual fine-tuning on language $t$, and let $S_t^m$ be the F1 score of the multi-lingual fine-tuning on language $t$. Assume there are $T$ fine-tuning languages, the average relative F1 score is defined as $R_T = \frac{1}{T} \sum_{t=1}^{T} \frac{S_t^m}{S_t^s}$.

Figure 4 demonstrates the change of fine-tuning performance along with the increasing number of source languages. From this figure, we can see that (1) the fine-tuning performance just slightly fluctuates when the number of source languages changes; (2) LF-MLF and Uniform-MLF do not evidently outperform single-lingual fine-tuning, and they have similar average relative F1 score over different number of source languages.

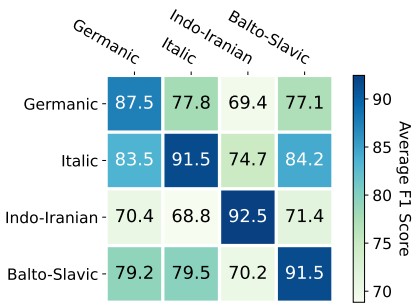
(a) Cross Language Branches

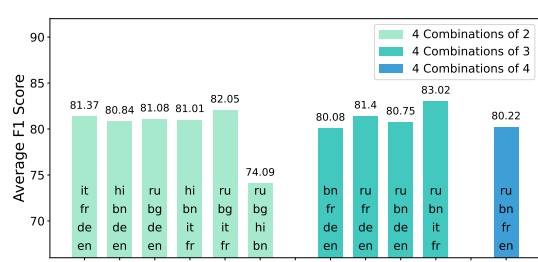
(b) Different Combinations of Language Families

Figure 5: Impact of the diversity on the fine-tuning languages' branches for LF-MLF.

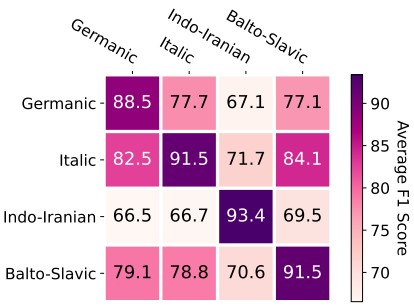
(a) Cross Language Branches

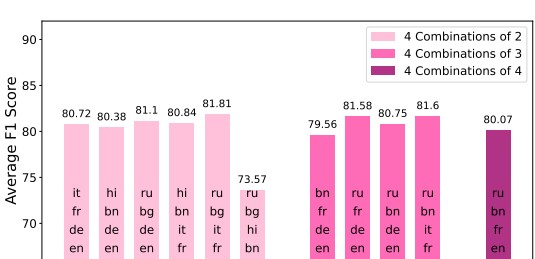
(b) Different Combinations of Language Families

Figure 6: Impact of the diversity on the fine-tuning languages' branches for Uniform-MLF.

| Language Family | Germanic | Italic | Indo-Iranian | Balto-Slavic |
|---|---|---|---|---|
| Fine-tuning Candidates | en, de | fr, it | bn, hi | bg, ru |
| Test Set for Zero-shot | af, nl | pt, es | mr, ur | pl, uk |

Table 7: Data allocation for the language branches.

## 5.4 Impact of the Diversity of Language Branches Covered by the Source Languages

After investigating the impact of the number of source languages, we further explore whether the diversity of language branches covered by the source languages impacts the zero-shot performance. In this section, we conduct experiments on the different branches of Indo-European language family based on the NER dataset. Specifically, we choose sixteen languages, which belong to four different language branches (Germanic, Italic, Indo-Iranian, Balto-Slavic) and each branch has four languages, from the NER dataset.

Firstly, we investigate the cross-language-branches transfer performance, namely fine-tuning on the four languages of a branch and observe the zero-shot performance on another branch, such as fine-tuning on Germanic and test zero-shot performance on Italic. The experimental results are illustrated in Fig. 5 (a) and Fig. 6 (a) for LF-MLF and Uniform-MLF respectively. In these figures, the vertical axis represents the source. These results present that: (1) the cross-language-branches transfer performance of each pair of branches is asymmetric; (2) LF-MLF has better cross-language-branches transfer ability than Uniform-MLF, which experimentally explains the superiority of LF-MLF and further verifies LF-MLF's less forgetting property.

Next, we explore the impact of language branches' diversity of the source languages. Eight languages are used as candidates for fine-tuning (i.e., source languages), while the other eight languages are used to test the zero-shot performance. The detailed allocation of these languages are reported in Table 7. Fixing the number of fine-tuning languages as four, we test the language branches' combinations that the source languages cover two branches, three branches and all four branches respectively, and the results are illustrated in Fig. 5 (b) and Fig. 6 (b). From these figures, we can see that:

(1) the language branches' diversity of the fine-tuning languages does not have a clear impact on the zero-shot performance, which means that covering more language branches might not lead to better zero-shot performance; (2) LF-MLF generally outperforms Uniform-MLF on most of the combinations.

## 6 Conclusion

This paper researches on a novel fine-tuning setting for multi-lingual language models, namely multi-lingual fine-tuning; furthermore, we propose a novel multi-lingual fine-tuning method, dubbed Less-forgetting Multi-lingual Fine-tuning (LF-MLF). Comparing with its mono-lingual fine-tuning counterparts, multi-lingual fine-tuning can achieve superior zero-shot performance; moreover, our proposed LF-MLF outperforms the baseline multi-lingual fine-tuning methods. Researching on mono-lingual fine-tuning provides several inspirations and has the potential to forge new trends in multi-lingual learning research. This paper only focuses on the multi-lingual fine-tuning scenario. In the future, we will extend our method to more general scenarios, such as less-forgetting multi-task learning.

## Acknowledgements

This work was supported in part by the National Key Research and Development Program of China under Grant No. 2021YFC3300303, the NSFC under Grants No. (62025206, 61972338, and 62102351), and Ningbo Science and Technology Special Projects of China under Grant No. 2021Z019.

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
