# OpenReview forum: "Less-forgetting Multi-lingual Fine-tuning"
_NeurIPS.cc/2022/Conference — NeurIPS 2022 Accept_

### Official Review · Reviewer_2LmV · 2022-06-26

**Rating:** 5
**Confidence:** 3
**Soundness:** 3 good
**Presentation:** 3 good
**Contribution:** 2 fair

**Summary:**

This paper presents a way of learning from multilingual annotated data. They focus on the zero-shot transfer to low-resource languages for tasks like NER, QA and NLI.  The authors claim that for this setting they need to optimize the model for two new objectives: one that minimizes the forgetting of the low-resource languages from pretraining, and one that ensures that when fine-tuning on the different multi-lingual annotated datasets that the descent direction between them is shared or is in common. They formulate this as a constrained optimization problem and call their method Less-forgetting Multi-lingual Fine-tuning (LF-MLF). They show consistent but not very large improved performance on NER, QA and NLI and look at where labelled data from more languages helps - and unsurprisingly it does.


**Questions:**

For Figure 3 and 4 please make sure that you specify if the x or y axis is the source or target.


**Limitations:**

This work does not discuss its limitations which is a limitation.


**Strengths And Weaknesses:**

Strengths

They show consistent improvements averaged over up to 45 zero-shot language directions over three different tasks which is quite extensive.

Weaknesses

It is not clear from the paper how their approach is different from the baseline methods. The authors should explain the most relevant baselines and describe how they are different: Project Conflicting Gradients method and Gradient Vaccine.  This is a major flaw in the paper.

They have limited the usefulness of the methods to a very narrow, specific use-case: multilingual and zero-shot. The less-forgetting direction could in theory be useful for monolingual fine-tuning as well as multilingual fine-tuning. The combined descent could be useful for transfer learning with labelled data too - not just zero-shot. Why did the authors not mention this or do any experiments along these lines?

It is probably incorrect to state line 75: "MTL methods are not suitable for multi-lingual fine-tuning" as the authors use such methods as baselines in the paper, even though they do not describe them and how they are different to the LF-MLF.

The authors do not discuss the size of labelled data for each of the multilingual cases or discuss how this affects the results. Eg. in Table 2 en, de, fr, ru etc. as performance on fine-tuning just on Russian gives better performance that just training on English which does not make sense.

The amount of improvement is not great. The simple uniform multilingual fine tuning method is the second best method and LF-MLF is ahead of it by only 0.5-1 point.

---

> ### Author Response · Authors · 2022-08-01
> **Response to Reviewer 2LmV**
>
> We would like to thank you for your insightful and invaluable comments. Our paper has been carefully revised according to the comments, and the revised version has been submitted to OpenView. The revised parts are highlighted in blue in our manuscript. Below is our point-to-point response.
>
> Q1. It is not clear from the paper how their approach is different from the baseline methods....
>
> A1. Thanks for the comments and we appreciate your carefulness. We have re-written the related work part for Multi-task Learning, where we have discussed the difference between  PCGrad [26], Gradient Vaccine [27] and our proposed LF-MLF  (see Section 2, line 63-73). The revised related work part for Multi-task Learning is presented as follows.
>
> "Multi-task Learning (MTL), which simultaneously learns multiple tasks, aims to achieve proper performance on all the including tasks. Recently, various MTL methods [14,15,16,17,18,19,20] have been proposed. Among them, PCGrad [19] and Gradient Vaccine [20] have achieved the state-of-the-art performance. Regarding the source languages of MLF as MTL tasks, MTL methods can be used in MLF. However, MTL methods just focus on the performance of including source languages (tasks), while they do not consider the performance of target languages. It will bring inferior zero-shot performance on the target languages. For example, PCGrad and Gradient Vaccine just focus on finding a proper common descent direction for the source languages, but they do not care about the target languages. Thus, they can not expect to achieve promising zero-shot performance. By contrast, the proposed LF-MLF finds the common descent direction that benefit the zero-shot performance...."
>
>
> Q2. They have limited the usefulness of the methods to a very narrow...?
>
> A2. Thanks for the insightful comments. This paper focuses on the multi-lingual fine-tuning problem, which is an emerging research topic in the area of Multilingual Language Models. The multi-lingual fine-tuning problem remains less-explored. Especially, the forgetting problem in multi-lingual fine-tuning is still unexplored. Researching on the less-forgetting multi-lingual fine-tuning is meaningful. Therefore, this paper focus on this meaningful topic.
>
> We highly agree your advice that our method has the potential to be used in a wide range of scenarios, e.g., transfer learning. It is an advantage of our method. In the future, we will adopt LF-MLF in other scenarios.
>
>
> Q3. It is probably incorrect to state line 75: "MTL methods are not suitable for multi-lingual fine-tuning" ....
>
> A3. Thanks very much for your advice. MTL methods only focus on the fine-tuning performance on the source languages, while they do not consider the zero-shot performance on the target languages. It makes them to have inferior zero-shot performance on the target languages. In the contrary, zero-shot performance cannot be ignored in multi-lingual fine-tuning (MLF). Therefore, although MTL methods can be used in MLF, they are not the very suitable. By contrast, LF-MLF focus on both the fine-tuning performance and zero-shot performance. Our experimental results verifies that LF-MLF has better zero-shot performance than the baseline MTL methods.
>
> We appreciate your comments very much. Our statement "MTL methods are not suitable for multi-lingual fine-tuning" seems too strong. To present more accurate and clear, we have re-written the related work part for Multi-task Learning, where we have discussed the difference between MTL and LF-MLF.
>
> Q4. The authors do not discuss the size of labelled data for each of the multilingual cases or discuss how this affects the results....
>
> A4. Thanks for your insightful comments. In this paper, we adopt the XTREME benchmark and keep its original settings unchanged (includes the size of labelled data) to be consistent with the previous works in the area of Multilingual Language Models. Furthermore, our work focuses on the multi-lingual fine-tuning (MLF) setting, where the compared MLF methods (Uniform-MLF, PCGrad-MLF, GradVac-MLF and LF-MLF) adopt the labelled data from all the source languages. Although the source languages may have different size of labelled data, the MLF methods have identical size of labelled data for they use the labelled data from all the source languages. Thus, our comparisons between the MLF methods are fair.
>
> Nevertheless, your advice is very valuable for us. Due to the time limitation of the rebuttal, we cannot conduct enough experiments to discuss the impacts bring by the he size of labelled data. However, in the later version, we will make our best effort to add sophisticated experimental analysis on the impacts bring by the size of labelled data.
>
> Q5. For Figure 3 and 4 please make sure that you specify if the x or y axis is the source or target.
>
> A5. Thanks for your valuable suggestions. In Figure 3 (a) and 4(a), the y axis is the source. We have specified it in our revised paper (please refer to the Section 5.4).

---

### Official Review · Reviewer_R8FH · 2022-07-04

**Rating:** 6
**Confidence:** 3
**Soundness:** 2 fair
**Presentation:** 3 good
**Contribution:** 3 good

**Summary:**

This paper proposes a method for maintaining multilinguality under fine-tuning when a set of languages is used for fine-tuning. The core idea is to maintain crosslingual generalization by finding updates that are deviating not too much from the original pretrained model and its loss, and those that many languages benefit from. The proposed loss is and its optimization are theoretically motivated and defined, and then evaluated on downstream tasks of the XTREME benchmark with XLM RoBERTa, where a few high-resource languages are chosen for fine-tuning, and the remaining languages are used for zero-shot evaluation.

[NOTE: presentation and overall score updated after author rebuttal]

**Questions:**

 [NOTE: all well addressed in revised version]
1. Please discuss [26] and [27] in Related Work as well, as baselines are based on them, and the differences should be discussed.
2. How are mini-batches represented in the loss definition? And how are they composed (mixed across languages or only one language? This might make a different for optimization.
3. How are the "heads" defined in Section 3.2? Why can one not use the pretraining head after having fine-tuned the rest of the parameters to get an estimate of L_p?
4. Figure 2 is not readable in black and white / grayscale.



**Limitations:**

Limitations are not addressed, except for that in some scenarios the proposed method does not outperform all other methods. Please discuss in which scenarios the method might not work, or where the assumption that being close to the original pretraining model/objective is not sufficient/desirable for downstream crosslingual generalization.

**Strengths And Weaknesses:**

Strengths:
1. The problem setting of zero-shot generalization *after* fine-tuning is novel and offers impactful applications.
2. The experimental results are largely convincing.
3. The detailed impact studies (Section 5.3 and following) are insightful and thorough.

Weaknesses: [NOTE: all well addressed in revised version]
1. The paper misrepresents multilingual fine-tuning ("multilingual fine-tuning, which has not been explored before" l.34; "the fine-tuning scenario where multiple source languages are involved in fine-tuning, namely multilingual fine-tuning, remains unexplored." l.59). It has been explored before, most prominently in the paper for the XTREME benchmark (https://arxiv.org/abs/2003.11080) that this very paper is also evaluating on. "translate-train" and "in-language multi-task" (from XTREME) yielded gains over monolingual fine-tuning, and should be discussed, in particular in relation to the Uniform MLF baseline presented in this work. Another work that builds on multilingual fine-tuning is for example https://arxiv.org/abs/2205.02022 (NAACL 2022). What hasn't been studied in this setting afaik is the problem of forgetting for some languages, fine-tuning was either done on all languages of interest or only on English.
2. The abstract and introduction are not well representing the focus of the remaining sections, as it does not introduce and motivate the forgetting problem, that the proposed method is mainly developed for and the title is derived from.
3. A comparison to all-language fine-tuning should be included as an upper bound, since training data is available for at least the NER task.
4. Significance tests should be performed, since some differences are sometimes small (and averages across multiple languages) and might not be strong enough to conclude superiority of one model over the other. At least standard deviation could be reported as well (since it's 3 runs).

---

> ### Author Response · Authors · 2022-08-01
> **Response to Reviewer R8FH**
>
> We would like to thank you for your insightful and invaluable comments. Our paper has been carefully revised according to the comments, and the revised version has been submitted to OpenView. The revised parts are highlighted in blue in our manuscript. Below is our point-to-point response.
>
> Q1. The paper misrepresents multilingual fine-tuning....
>
> A1. Thanks for the comments and we appreciate your carefulness. We have fixed the misrepresentations in our revised paper. Specifically, the misrepresentations are re-written as “Few works [8,9] have investigated the multi-lingual fine-tuning setting, but they do not consider the forgetting of MLLMs' cross-lingual generalization ability. To fill this gap, this paper focuses on the forgetting problem in multi-lingual fine-tuning”. Furthermore, we have re-organized and re-written the abstract and introduction in our revised paper. In the revised version, we have made our best to clearly claim our contributions. We are very grateful for your patience to read our revised paper.
>
> Q2. The abstract and introduction are not well representing...
>
> A2. Thanks for the insightful comments. We have re-organized and re-written the abstract and introduction in our revised paper. According to your advice, in the revised version, we have introduced and motivated the forgetting problem by adding more details. We are very grateful for your patience to read our revised paper.
>
> Q3. A comparison to all-language fine-tuning should be included as an upper bound, since training data is available for at least the NER task.
>
> A3. Thanks very much for your advice. We have added a comparison to the all-language fine-tuning (ALL-MLF). Due to the page limit, we have put the comparison in the Appendix section of our revised paper. The results can be found in the Table 8, 9, 10, 11, 12, 13 of our revised paper. Without any doubt, ALL-MLF has superior zero-shout and fine-tuning performance than LF-MLF and the baseline methods. ALL-MLF is the upper bound for MLF methods. We will make effort to approach the upper bound in the future.
>
> Q4. Significance tests should be performed...
>
> A4. Thanks for your advice. In the revision, we have reported the standard devisations of all results in Table 1, 2, 3, 4, 5, 6. Furthermore, for the zero-shot performance results (Table 1, 2, 3), the pairwise t-tests at 0.05 significance level are conducted. The t-tests demonstrates that LF-MLF is statistically superior to most of the baselines on each task. Besides, to comprehensively evaluate the superiority of LF-MLF, we further utilized Friedman test and Bonferroni-Dunn test to verify the superiority of LF-MLF. We can see that LF-MTF achieves highly superior results to other baseline methods  (Please refer to the Section 5.2.1 line 208-222 of our revised paper).
>
> Q5. Please discuss [26] and [27] in Related Work as well...
>
> A5. Thanks very much for your advice. We have re-written the related work part for Multi-task Learning, where we have discussed the difference between  PCGrad [26], Gradient Vaccine [27] and our proposed LF-MLF.  We are very grateful for your patience to read our revised paper (see Section 2, line 63-73).
>
>
> Q6. How are mini-batches represented in the loss definition? And how are they composed...?
>
> A6. Thanks very much for your comments. In the loss definition, each language has a language-specific mini-batch. We sum losses over these language-specific mini-batches together and then conduct backward propagation. In the other word, each mini-batch is composed of only one language.
>
> Q7. How are the "heads" defined in Section 3.2? Why can one not use the pretraining head after having fine-tuned the rest of the parameters to get an estimate of L_p?
>
> A7. Thanks very much for your comments. In Section 3.2, the head represents the output layer for specific tasks. For example, a masked language modeling head represents the output layer for predicting the masked words, and a token classification head represents a token-level output layer for classification.
>
> We do not mean “one can not use the pretraining head after having fine-tuned the rest of the parameters to get an estimate of L_p?” Instead, We mean that “one can not   --directly--  use the pretraining head after having fine-tuned the rest of the parameters to get an estimate of L_p?” This is because, in the fine-tuning phase, if we want to estimate the L_p, we have to replace the down-stream task-specific output layer with the masked word  prediction layer. Therefore, we argue that “one can not directly use the pretraining head after having fine-tuned the rest of the parameters to get an estimate of L_p”
>
>
> Q8. Figure 2 is not readable in black and white / grayscale.
>
> A8. Thanks very much for your comments. We have changed line style of Uniform-MLF to dash in our revised paper, which enable this figure to be readable in black and white / grayscale.

---

> > ### Comment · Reviewer_R8FH · 2022-08-08
> > **Weaknesses well addressed**
> >
> > Thank you for addressing the listed weaknesses and questions. I read the revised parts of the paper and find them well addressed. A discussion of limitations remains missing nevertheless.

---

> > > ### Author Response · Authors · 2022-08-09
> > > **Sincere thanks and complement of the limitations section**
> > >
> > > We highly appreciate your time for reading our revised paper. Your constructive and professional comments help a lot for improving this work. As to the limitations, we have discussed from two perspectives: (1) the assumption of being close to the original pretraining benefits the cross lingual generalization; (2) our theoretical analysis on the Pareto stationarity. We have added it in the current revised version (see Section A.1) as follows.
> > >
> > > 1. Assumption. In Theorem 1, we assume that $L_p(\theta_f)$ can be approximated by its second order Taylor expansion. This assumption is widely adopted [21] [22] [23] and can ensure proper performance over various deep neural network models and learning scenarios. With this assumption, we obtain an upper bound for the forgetting of multi-lingual fine-tuning, which demonstrates that being close to the original pretraining model can reduce forgetting, in other words, enhance cross-lingual generalization. Thus, if the assumption of second order Taylor approximation failed, being close to the original pretraining model/objective is not sufficient/desirable for downstream cross-lingual generalization. Note that relaxing this assumption remains an open problem, and if this assumption can be relaxed, our proposed methods can be used in more widely models and scenarios.
> > >
> > > 2. Theory. In our theoretical analysis, we prove that our method can achieve a Pareto stationary point. Pareto stationarity is only a necessary condition for achieving Pareto optimality. Theoretical analysis of the gap between Pareto stationarity and Pareto optimality has not been studied in this work. Furthermore, in the setting of deep learning, the gap between Pareto stationarity and Pareto optimality remains less explored in both the area of multi-objective optimization and machine learning. In the future, we will explore this challenging problem.

---

### Official Review · Reviewer_7aAk · 2022-07-11

**Rating:** 6
**Confidence:** 4
**Soundness:** 4 excellent
**Presentation:** 3 good
**Contribution:** 3 good

**Summary:**

The paper proposes an investigation of how to fine-tune a model with multilingual data effectively with a theoretical and empirical evidence. They proposed a method to avoid catastrophic forgetting by using interior-point solver to optimize the objective function based on the theoretical work. The authors claim that the method is able to reach the Pareto stationary -- no more changes occurred by optimizing multiple objectives by applying optimization. They show the superiority of the approaches in Named Entity Recognition, Question Answering and Natural Language Inference, and several languages.


**Questions:**

- Did you run any significant tests or compute the std of all results (for all runs)? The scores for LF-MLF are similar to Uniform-MLF for some tasks and settings.
- I am curious about the running time of the optimization. Does this method efficient in practice?

**Limitations:**

There is no specific limitation section provided in the paper.

**Strengths And Weaknesses:**

Strengths:
- Interesting work with a good theoretical foundation
- The task is very useful for learning multilingual models while optimizing the model to reach closer to the upper bound

Weaknesses:
- The paper is not easy to follow. The paper method is unclear until I carefully read the paper several times. I would say the paper would be easier to follow if they put more details on the Algorithm and the introduction. For example, what method does the paper use to optimize the weights, not just pointing to the equation. And, it would be great to add a short description of the method in the introduction so that the first readers can understand the proposed approach.
- The method is not well-motivated with a clear hypothesis. Therefore, I cannot find the rationale why the LF-MLF is useful in the abstract and introduction.

Typo:
- In conclusion, Learning => learning

---

> ### Author Response · Authors · 2022-08-01
> **Response to Reviewer 7aAk**
>
> We would like to thank you for your insightful and invaluable comments. Our paper has been carefully revised according to the comments, and the revised version has been submitted to OpenView. The revised parts are highlighted in blue in our manuscript. Below is our point-to-point response.
>
> Q1. The paper is not easy to follow. The paper method is unclear until I carefully read the paper several times. I would say the paper would be easier to follow if they put more details on the Algorithm and the introduction....
>
> A1. We thank the reviewer for the patience and valuable suggestions. We have carefully revisited the writing issues of our paper and try our best to improve the presentation. To present our proposed method more clear, we have revised our paper from the following two aspects.
>
> (1) To explain the motivation and mechanism of our method, we have re-organized and re-written the abstract and introduction of our paper. For example, in the revised abstract, we have explained our method from a broad perspective: “In LF-MLF, we cast multi-lingual fine-tuning as a constrained optimization problem, where the optimization objective is to minimize forgetting, and constraints are reducing the fine-tuning loss. ”. Correspondingly, in the revised introduction, we have concluded the mechanism of our method as: “This paper proposes to find a less-forgetting descent direction, which prevents a MLLM from forgetting the cross-lingual generalization ability and is a common descent direction for the source languages. To find this less-forgetting descent direction, we cast MLF as a constrained optimization problem. The optimization objective is to minimize the forgetting of a MLLM's cross-lingual generalization ability, and the constraint is that the direction should be a descent direction common to all the source languages.”.
>
> (2) We have added a toy example to illustrate the mechanism of our proposed LF-MLF method in Section 3.3 of our revised paper. In the toy example, we explain the  mechanism and advantage of LF-MLF from a geometric view.
>
> Q2. The method is not well-motivated with a clear hypothesis. Therefore, I cannot find the rationale why the LF-MLF is useful in the abstract and introduction.
>
> A2. Thank you for your suggestions to improve the presentation of this paper. In response to advice, to better explain the rationale of LF-MLF’s effectiveness, we have (1) added more background knowledge and insights in the abstract and introduction of our revised paper; (2) we have explain the rationale of LF-MLF from a geometric view by giving a toy example in Section 3.3 of our revised paper.
>
>
> Q3. Did you run any significant tests or compute the std of all results (for all runs)? The scores for LF-MLF are similar to Uniform-MLF for some tasks and settings.
>
> A3. Excellent comments. In the revision, we have reported the standard devisions of all results in Table 1, 2, 3, 4, 5, 6. Furthermore, for the zero-shot performance results (Table 1, 2, 3). The pairwise t-tests at 0.05 significance level are conducted. The t-tests demonstrates that LF-MLF is statistically superior to most of the baselines on each task.
>
> Besides, to comprehensively evaluate the superiority of LF-MLF, we have further utilized Friedman test as the statistical test to analyze the relative performance among the compared methods across the three applications. At 0.05 significance level, the Friedman statistics is 5.5, and critical is 4.76.  Thus, at 0.05 significance level, the null hypothesis of indistinguishable performance of LF-MTF among all compared methods is clearly rejected. Subsequently, we employ the Bonferroni-Dunn test as the post-hoc test by regarding LF-MLF as the control approach. Figure 2 (please refer to the Figure 2 in the revised version of our paper) reports the CD diagrams at 0.1 significance level, where the average ranks of the compared approaches is marked along the axis. From this figure, we can see that LF-MTF achieves highly superior results to other compared baseline methods.
>
> Q4. I am curious about the running time of the optimization. Does this method efficient in practice?
>
> A4. Thanks for your insightful comments. The running time of optimizing Problem 2 can be concluded as $T(T+1) C_1 +  C_2(T)$, where $T$ is the number of the source languages, $C_1$ is the time of computing the inner product between gradients $\nabla L(\theta_{k-1})^{\top} \nabla L(\theta_{k-1})$, and $C_2(T)$ is the time of solving a $T$-dimensional quadratic programming problem. The quadratic programming problem  is solved by using the solvers.qp() function of cvxopt package. We test the running time on the platform which has a NVIDIA Tesla V100 32GB GPU and a Intel(R) Xeon(R) Silver 4216 CPU @ 2.10GHz. By repeated testing 100 times, the average $C_1$ is 1.28 ms, and  $C_2(T)$ is approximately $50+10*T$ ms. When $T =10$, the running time of optimizing Problem 2 is 378ms. It is efficient and affordable in the training phase.

---

> > ### Comment · Reviewer_7aAk · 2022-08-09
> > **Authors addressed the issue**
> >
> > Thanks for addressing the issue and answering my questions. Now, the paper is clearer than before. And the method is well-motivated, and I can find the significance test in the revised version.

---

> > > ### Author Response · Authors · 2022-08-09
> > > **Thank you for your reply**
> > >
> > > Thank you for your reply, and we appreciate your time for reading our revised paper. Your comments help us a lot for improving our work.

---

### Meta-Review · Area_Chair_aUP8 · 2022-08-26

**Recommendation:** Accept
**Confidence:** Less certain

**Metareview:**

The paper proposes a method for finetuning multi-lingual pre-trained language in multiple languages simultaneously.  The task is formalized as a  constrained optimization problem and the upper bound of the forgetting is given in theory.  A method is developed for multi-lingual fine-tuning to minimize the upper bound.  Experiments are conducted in multiple downstream tasks and the model is fine-tuned in a few high-resource languages and the performance is improved in low resource languages as zero-shot settings.  The authors responded the reviewers' concerns and the reviewers agree the responses addressed their concerns.  The paper is recommended to be accepted, and I ask the authors to carefully prepare the final camera-ready version based on the reviewers' feedback.

**Award:**

No

---

### Decision · Program_Chairs · 2022-09-14

Accept